# Identification of Shrinkage and Growth Patterns of a Shrinking City in China Based on Nighttime Light Data: A Case Study of Yichun

**Ying Zhou [1], Chenggu Li [1,\*], Zuopeng Ma [2], Shuju Hu [1], Jing Zhang [1] and Wei Liu [1]**

[1] School of Geographical Sciences, Northeast Normal University, Changchun 130024, China;
zhouy751@nenu.edu.cn (Y.Z.); husj163@nenu.edu.cn (S.H.); zhangj888@nenu.edu.cn (J.Z.);
liuw445@nenu.edu.cn (W.L.)
[2] Northeast Institute of Geography and Agroecology, Chines Academy of Science, Changchun 130102, China;
mazuopeng@iga.ac.cn
\* Correspondence: lcg6010@nenu.edu.cn

**Abstract:** Urban shrinkage has become a topic of major concern to scholars of geography and urban science. However, the methods of identifying urban shrinkage and growth have mostly focused on traditional statistical methods, and studies based on nighttime light (NTL) data are rare. Here, we use the NTL data for 56 months from 2012 to 2019 obtained by the Visible Infrared Imaging Radiometer Suite (VIIRS) on board the Suomi National Polar Orbiting Partnership (NPP) to identify the shrinkage and growth patterns of Yichun in China, by calculating the slope of the NTL radiance value after denoising. At the same time, by combining high-resolution Google satellite images and traditional demographic data, we analyzed the shrinkage characteristics of Yichun. The results of the study confirmed the characteristics of partial shrinkage in China's shrinking cities. In addition, the use of NPP-VIIRS NTL data was able to more accurately identify the urban shrinkage and growth patterns, and may also be seen to present a more objective picture of reality, thus providing a new perspective for studies of urban shrinkage.

**Keywords:** urban shrinkage; NPP-VIIRS; spatial pattern; Yichun

---

## 1. Introduction

The terminology of "urban shrinkage" originates from "Schrumpfende Städte", referring to the phenomenon of population decline and economic recession caused by deindustrialization in Germany [1]. It is not a new phenomenon [2] and has been spreading worldwide [3–8]. Since the mid 20th century, with the adjustment of global economic structure and the ageing of the population, more than 25% of the world's major cities have been considered to be shrinking cities [9]. Urban shrinkage has become a topic of major concern to scholars of geography and urban science. Due to the great heterogeneity of political, economic, geographic, and social backgrounds across different cities, the causes and spatial characteristics of urban shrinkage have varied by countries in different national contexts [10]. In many Western countries, urban shrinkage is mostly caused by deindustrialization [11] and suburbanization [10], often resulting in massive population loss and severe economic recessions [12,13]. In terms of space, such shrinkage is characterized by a comprehensive recession in the central area of the city, the emergence of a large number of vacant residential and office buildings, and the oversupply of urban service facilities [14]. Examples of cities where this has taken place include Detroit, Cleveland, and Youngstown in the United States, Leipzig in Germany, Manchester in the United Kingdom, and many mining cities in Australia [15]. As far as commonality is concerned, many of China's resource-based cities are also facing a chain reaction process, such as the depletion of resources, industrial recession, reduction of

jobs, population loss, and economic recession. In terms of specific characteristics, the shrinkage caused by de-industrialization, and by the growth brought about by urbanization and re-industrialization, overlap in China. Many scholars hold the view that China's shrinking cities are mainly characterized by the scattered and fragmented shrinkage of partial areas [16,17], rather than an overall recession. As such, it is necessary to refine the current understanding of the internal shrinkage pattern of China's shrinking cities.

At present, while the research on urban shrinkage both in China and internationally is on the rise, the definition of the shrinking city has been inconsistent [18], leading to a dilemma of identification. Generally, scholars believe that population decline is the main sign of urban shrinkage [19–24], thus the identification of shrinking cities mainly relies on population data [20–24]. However, traditional population data has the limitations of collection difficulties, inaccurate data statistics, statistical standard changes, long re-update cycles, and rough spatial expressions [25,26], among other limitations, thus leading to erroneous conclusions. Faced with the measurement dilemma, some scholars began to use nighttime light (NTL) composite data obtained by satellites to identify shrinking cities [27–30]. In urban space science research, the NTL data obtained by the OLS (Operational Linescan System) sensor under the Defense Meteorological Satellite Program (DMSP) is most commonly used. These data provide unified, spatially explicit, continuous, and timely annual, cloud-free, composited and stable light data. Therefore, they have been widely used in evaluating the process of urbanization [28], economic development dynamics [31–33], population distribution [34,35], power consumption [36], drawing poverty maps [37], and mapping out urban areas [38–40]. However, DMSP-OLS NTL data still have some defects, such as oversaturation, low spatial resolution (about 1 km at the equator), a lack of on-board correction and satellite correction, and a blooming effect on surrounding areas [41–44]. Moreover, the National Geophysical Data Center (NGDC) of the USA unfortunately stopped producing monthly composites of DMSP/OLS after February 2014. Its successor, the Visible Infrared Imaging Radiometer Suite (VIIRS) on board the Suomi National Polar Orbiting Partnership (NPP), launched in October 2011, and these data have been systematically updated for the period starting in April 2012. The NPP-VIIRS NTL data make up for the defect of the DMSP-OLS data, as they have a higher spatial resolution (375 m and 750 m at nadir), with a spectral range of 500–900 nm that is sensitive to very low visible light. In addition, an onboard calibration system is utilized to improve the quality of the NPP-VIIRS NTL data. Hillger et al. argue that the NTL data obtained from VIIRS provide more abundant information about human settlement and economic activities than DMSP NTL data [45]. A former study has, furthermore, demonstrated its superiority over DMSP-OLS with regard to urban characterization [45,46].

To date, urban studies, using NPP-VIIRS NTL data, have focused on the following areas: (1) the relationships between the NPP-VIIRS NTL data and socioeconomic indicators, such as population, GDP, and house vacancy rate [25,36,47,48]; (2) extracting built-up areas and dynamically monitoring urban expansion by using the NPP-VIIRS NTL data [36,49,50]; (3) exploring the spatial distribution of the population using NTL data [46,51]. These studies have provided insights and foundations for using NPP-VIIRS NTL data to identify urban shrinkage and growth patterns. Firstly, many studies have shown that NTL radiance values have a significant statistical relationship with gross domestic product (GDP) and population, which illustrates the comprehensiveness of NTL data and its strong relevance to human activities. In addition, the capacity of NTL time-series data to dynamically detect changes in urban landscapes demonstrates the objectivity and timeliness of these data. Finally, using NPP-VIIRS NTL data to spatialize socio-economic indicators can be seen as being more reliable than using traditional data. In order to test the superiority of NPP-VIIRS NTL data, some scholars have made meaningful attempts to contribute to research on city shrinkage. At present, there are two methods of processing the NTL data for urban shrinkage research. Firstly, Du et al. (2017) and Liu et al. (2018) employed a calculation of the difference of different years' NTL radiance value in every grid to identify urban shrinkage and growth; however, this difference method cannot determine the continuity and trend of urban shrinkage and growth. Secondly, Li et al. (2019) used the NPP/VIIRS NTL data in

the calculation of the change slope of each grid's NTL radiance value to identify the urban shrinkage and growth pattern; however, the data here were not carefully processed through the implementation of noise removal. Moreover, the scope of the research focused on the whole prefecture-level city rather than the urban area, which differs from the western cities where there are urbanized areas within administrative boundaries.

Based on the above analysis, this paper takes Yichun, a typical resource-based city in China, as an example, using the NPP-VIIRS nighttime light (NTL) data of 56 months from 2012 to 2019 to identify the shrinkage and growth pattern. The research scope focuses on the built-up area in the urban area of Yichun, for which the area was extracted by the visual interpretation of a high-resolution Google satellite image. The key aim of the research was to calculate the slope of the change of the NTL radiance value of each grid in the built-up area in order to then identify the growth and shrinkage of the city. The main objectives of this paper are as follows: to implement an objective and effective method to understand the shrinkage and growth pattern of Yichun; to summarize the shrinking characteristics and types of Yichun City; and to explore the advantages of using NPP-VIIRS NTL data for urban shrinkage identification. Based on the body of research of typical cases in China, this paper hopes to provide a new perspective for objectively and meticulously identifying shrinkage and growth patterns within a single city, thus providing a basis for the governance of resource-based cities.

## 2. Study Area and Background

In recent years, an increasing number of Chinese scholars has paid attention to shrinking cities, with a large number of studies having identified the situation and distribution of urban shrinkage in China [20,52,53]. These studies all indicate that urban shrinkage is common in China. For example, Long et al. (2015) used the 2000 and 2010 census data to define 180 shrinking cities in China; Wu et al. (2015) used the urban population data in 2007 and 2016 to identify 80 shrinking cities in China; Zhang et al. (2016) and Wu et al. (2018) found over a third of cities in China to be experiencing different degrees of population shrinkage at the county scale. Moreover, these results all showed that many cities in northeast China are experiencing the most serious shrinkage, and most of them are resource-based cities.

Yichun City is located in the northeast of China (Figure 1) and is China's largest resource-based city, with the development of forest resources being its leading industry. The development of forest areas in Yichun began in 1948. Around the development of local forest resources, large-scale pillar industry systems such as forest management, timber production, and wood processing were gradually formed. However, given the long-term, high-load logging, Yichun now faces the dilemma of the depletion of forest resources and industrial decline. On March 17, 2007, the Chinese government confirmed the city to be one of the first 12 resource-exhausted cities in the country, and regarded it as the first pilot project for the economic transformation of forest resource-based cities. In 2011–2013, Yichun stopped its major logging and its annual timber production was reduced in one step, leaving only 330,000 cubic meters of logging to continue. Commercial harvesting was completely stopped in December 2013. Since then, the economy and population of Yichun City have been severely affected. The total population of Yichun's urban area dropped from 795,632 in 2012 to 714,269 in 2018, with a total loss of 81,363. The city's GDP fell to its lowest point in 2015 (Figure 2), and the proportion of its secondary industry also fell from 32.6% to 21.64%. According to the Shrinking Cities International Research Network (SCIRN), the definition of a shrinking city is that of a densely populated urban area with a minimum population of 10,000 residents that has faced population losses in large parts for more than two years and is undergoing economic transformations, with some symptoms of a structural crisis [12]. In recent years, Yichun City has faced the absolute loss of population and the crisis of a severe economic recession. As such, Yichun may be taken as a typical city for the purposes of studying the urban shrinkage of China's cities.

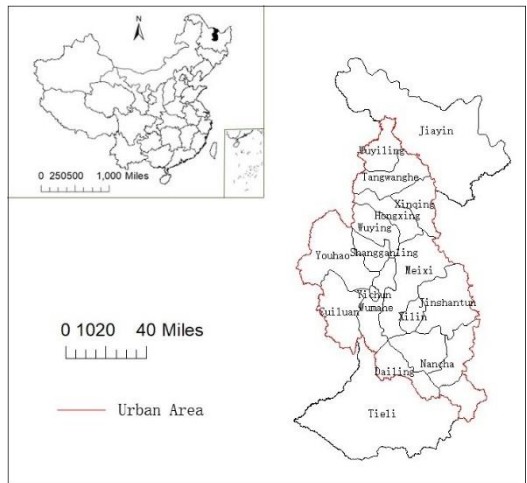

**Figure 1.** Location of Yichun in China.

Yichun City has jurisdiction over 15 municipal districts, and every district has jurisdiction over townships, towns, streets, and several forest farms. Generally speaking, forest farms have the characteristics of a large space, but small internal built-up areas. For the convenience of analysis, the streets with connected areas and high building density in each district are here collectively referred to as core blocks, while the streets with low building density, and the townships and towns that are not connected to the built-up area of core blocks, are collectively referred to as secondary blocks; the remainder are forest farms.

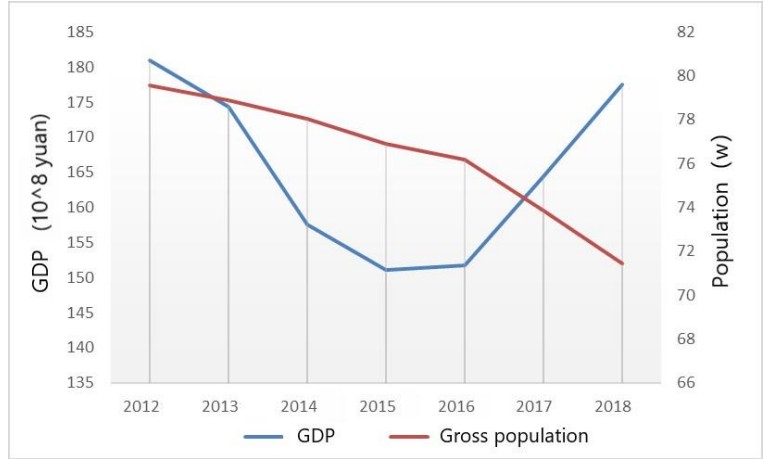

**Figure 2.** Total population and GDP trends of downtown Yichun in 2012–2018.

## 3. Data Processing and Methods

### 3.1. Data Source

A global composite of NPP-VIIRS NTL images from 2012 to 2019 were provided by the Earth Observation Group at the National Geophysical Data Center of the National Oceanic and Atmospheric Administration (downloaded from the website http://ngdc.noaa.gov/eog/viirs/download momthly.html). The current version 1 VIIRS DNB composites contain two different configurations, that is, "VCM"(VIIRS Cloud Mask) and "VCMSL" (VIIRS Cloud Mask Stray Light). The former excludes any data contaminated by stray light (typically solar illumination) [54], but the time period covered is longer. The latter data, impacted by stray light, were corrected, not removed, and thus contained greater coverage of high-latitude regions but of worse quality. In order to explore the urban shrinkage and growth pattern of Yichun since the banning of logging in 2011, this study selected

the "VCM" data covering a wider time range in order to obtain a higher estimation accuracy. Due to the high stray light images of high latitude areas in the summer [55], the seriously influenced NTL data were directly excluded as invalid data (including the months of May, June, July, and August). Therefore, the data used in this study included the NPP-VIIRS NTL data of 56 months from 2012 to 2019, with a spatial resolution of 413 m.

Google Earth satellite imagery was downloaded from the 91 map assistant, with an image level of 18 and a high spatial resolution of 0.4 m. For the purposes of this study, this satellite imagery was used as ancillary information to determine the boundaries of the built-up area of the city by visual interpretation, thereby extracting NTL data within the built-up area. Following this, an overlay analysis of Yichun's shrinkage characteristics was applied. The demographic and economic statistics used in this study were taken from the Heilongjiang Statistical Yearbook, the Yichun Statistical Yearbook, and the Yichun City Statistical Report on National Economic and Social Development.

### 3.2. Data Pre-Processing

Given the characteristics of NPP-VIIRS time series images and the needs of specific research problems, the images need to be corrected prior to undertaking the analysis (Figure 3). With on-board correction and satellite correction, the inter-annual systematic geolocation shift of NPP-VIIRS NTL data is small; for this reason, so we assumed that the monthly data for 2012–2019 comprised a coherent dataset. We then smoothed extreme values from the image by determining the optimal threshold. In addition, due to the characteristics of the artificial nighttime radiation source and the sensor itself, the existence of the temporary light source and the blooming effect are two important phenomena to consider when using NPP-VIIRS data. To rectify these defects, we attempted to perform specific processing by using high-resolution Google satellite imagery.

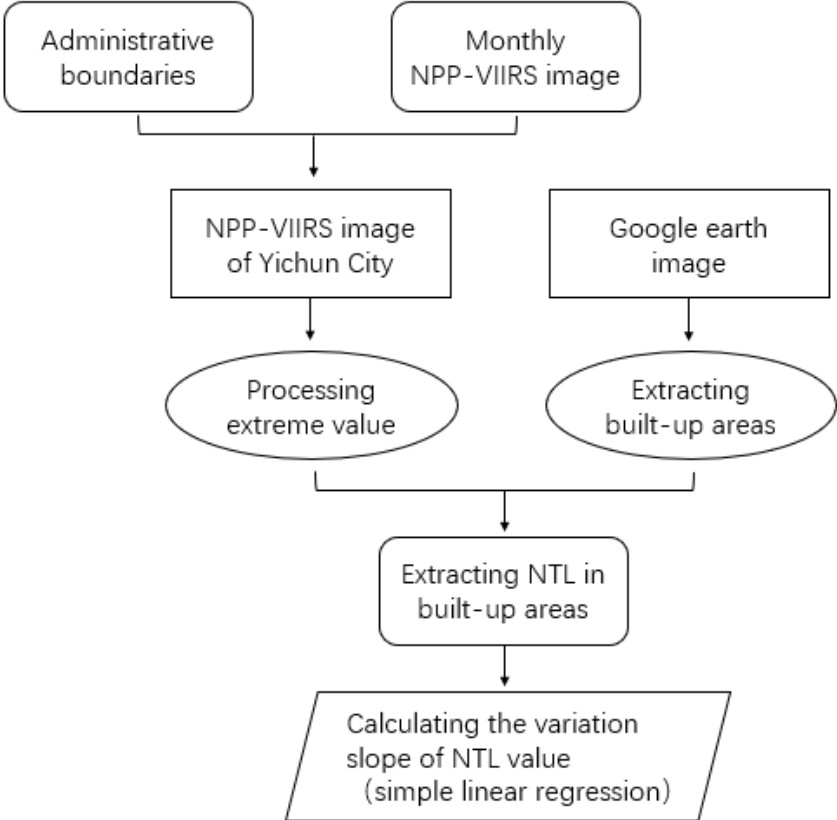

**Figure 3.** Flowchart of National Polar Orbiting Partnership (NPP)-Visible Infrared Imaging Radiometer Suite (VIIRS) nighttime light (NTL) data processing.

### 3.2.1. Extreme Value Processing

Firstly, the NPP-VIIRS NTL data were converted into a horizontal axis Mercator (UTM) projection with reference to the WGS84 datum in an ArcGIS environment. Then, the NTL image of downtown of Yichun was extracted by superimposing this with the vector data of Yichun City. Temporal lights, such as those from fires, explosions, and other background noises, were not filtered out in the monthly composites [56,57]. Yichun and Harbin are the prefecture-level cities of Heilongjiang Province. Harbin is the capital of Heilongjiang Province and its most developed city, meaning that the pixel values of the other areas could not, theoretically, exceed the highest value of NTL in this city. Harbin's highest NTL radiance value was 153 nanoWatts/cm2/sr, which was used as a threshold to correct outliers. As there was no description about those pixels in the metadata of the original NPP-VIIRS data, we assumed that the negative DN values of those pixels were caused by background noise and outliers from data processing, and removed the noise by setting their pixel values to 0. And the extreme values in the data were de-noised as follows:

$$DN_{(n,i)} = \begin{cases} DN_{(n,i)} = DN_{(n,k)}, DN_{(n,i)} > 153 \\ DN_{(n,i)} = DN_{(n,i)}, 0 \leq DN_{(n,i)} \leq 153 \\ DN_{(n,i)} = 0, DN_{(n,i)} < 0 \end{cases} \tag{1}$$

where $DN_{(n,i)}$ denotes the radiation value of i th pixel in n th month, $DN_{(n,k)}$ denotes the maximum radiation value of eight pixels directly adjacent to the i th pixel ($DN_{(n,k)} \leq 153$, If the adjacent pixel values were all greater than 153, then the maximum value of the eight neighbors of each pixel in the immediate eight-neighbor area were selected). After this process, all of the pixel values in the corrected NPP-VIIRS data were lower than 153 and higher than 0.

### 3.2.2. Extraction of NTL Images of Built-Up Areas

The NPP-VIIRS sensor is highly sensitive to very low levels of visible light and can significantly improve the detection ability of anthropogenic lighting from cities, ships, and oil flares at night [44,57]. However, its light blooming effect and the influence of temporary light cannot be ignored. This study used high-resolution satellite images to extract the NTL of built-up urban areas, excluding the impact of changes in the radiance value of non-urban built-up areas on the identification of urban shrinkage and growth patterns.

First, the urban built-up area was manually divided into a set of vector polygons by visual interpretation using the latest Google Earth satellite imagery, with high resolution in the ArcGIS environment. Next, the annual average NTL image and each grid's slope of the variation of the radiance value of the pre-processed NTL image were calculated. The latter results were then resampled to a grid map of 100 m * 100 m in order to make the scattered and small built-up area polygons more accurately extract the corresponding range of NTL images. Following this, the built-up polygon was employed to perform mask extraction and, finally, the annual average NTL image and the slope image of the NTL changes within the built-up area were obtained.

### 3.3. Method

### 3.3.1. Simple Linear Regression

Regression analysis is a statistical analysis method that determines the quantitative relationship between two or more variables. In the regression analysis, only one independent variable and one dependent variable are included, and the relationship between the two can be represented by straight line approximation. This regression analysis is called a simple linear regression analysis. In this paper,

taking the time as the x-axis and the monthly NTL radiance value as the y-axis, a linear regression was performed for each grid during the 56 months from 2012 to 2019 (Equations (2)–(4)):

$$y = ax_i + b + \varepsilon_i \tag{2}$$

$$a = \frac{\sum_{i=1}^{n}(x_i - \overline{x})(y_i - \overline{y})}{\sum_{i=1}^{n}(x_i - \overline{x})} \tag{3}$$

$$\overline{x} = \frac{1}{n}\sum_{i=1}^{n}x_i, \; \overline{y} = \frac{1}{n}\sum_{i=1}^{n}y_i \tag{4}$$

where $i$ denotes the month, $a$ denotes the slope changes of NPP-VIIRS NTL radiation value, $b$ denotes the intercept of the NTL radiation value, and $\varepsilon_i$ denotes random error. We use the positive and negative regression coefficients $a$ to characterize the trend of nighttime radiation values, and the absolute value of the regression coefficients to quantify the trend of nighttime radiation values. If the value of a is greater than 0, it indicates an upward trend in the time series, whereas if the value is less than 0, it indicates a downward trend.

### 3.3.2. Metric Method

Through regression analysis, we obtained the slope of the radiance variation of each grid, following which we discerned the city's shrinkage according to the slope value. The metric at the 100 m * 100 m grid scale was divided into five categories: significant growth (a ≥ 0.30), slight increase (0.01 < a < 0.30), stable (−0.01 ≤ a ≤ 0.01), light shrinkage (−0.3 < a < −0.01), and significant shrinkage (a ≤ −0.3). By using the "Zonal Statistic" tool in ArcGIS, we calculated the average nighttime change slope of each district and each sub-district, following which the area was divided into four categories: growth (a > 0.03), stable (0 ≤ a ≤ 0.03), shrinkage (−0.1 ≤ a < 0), and significant shrinkage (a ≤ −0.1).

## 4. Temporal and Spatial Patterns of Shrinkage and Growth in Yichun City

*4.1. Overall Temporal and Spatial Pattern of Shrinkage snd Growth*

### 4.1.1. Regional Differences in Urban Shrinkage

The NTL data showed the sum of the slopes of all grids in the urban area of Yichun City from 2012 to 2019 to be −222.76 nanoWatts/cm2/sr, indicating the city's shrinkage to be very significant. Using the "Zonal Statistic" to quantify the mean value of the slope of the NTL radiance value to each district (Figure 4), it was found that the development of each district was uneven, and that shrinkage and growth co-exist in Yichun. Among the 15 districts in Yichun City, nine districts were found to have developing stagnation, accounting for 60%; three districts were found to be in a state of shrinkage, accounting for 20% and were mainly distributed in the central area around Yichun District; three districts were found to be in a state of overall growth, accounting for 20% and with a relatively scattered distribution.

The districts that have grown steadily are Tangwanghe District, Cuiluan District, and Dailing District, located in the northern, central, and southern parts of Yichun City, respectively. The growth rate of these three districts did not emerge as significant. The forest resources in Tangwanghe District are relatively abundant, with the average slope of the monthly average NTL radiance value demonstrating the growth rate of this district to be the most marked, with a value of 0.057 nanoWatts/cm2/sr. The Dailing district had the second highest rate of growth, at 0.056 nanoWatts/cm2/sr, while the growth rate of Cuiluan District was relatively small, at 0.031 nanoWatts/cm2/sr. In the three districts manifesting shrinkage, Yichun District, the economic center of Yichun City, was found to have the most severe shrinkage. The average monthly mean light fluctuation slope of the whole district emerged as −0.238 nanoWatts/cm2/sr. The population change data of Yichun District also provided evidence of the shrinkage of the district. From 2012 to 2017, the total population of Yichun District decreased from 162,066 to 15,850; in particular, after 2014

the population fell rapidly, and the annual population loss exceeded 1000. The overall shrinkage degrees of Wumahe district and Xilin district were found to be relatively light, with average monthly variation NTL slopes of −0.011 nanoWatts/cm2/sr and −0.014 nanoWatts/cm2/sr, respectively. From 2012 to 2017, the population of Wumahe District and Xilin District was seriously depleted, with net outflows of 3000 and 5740, respectively.

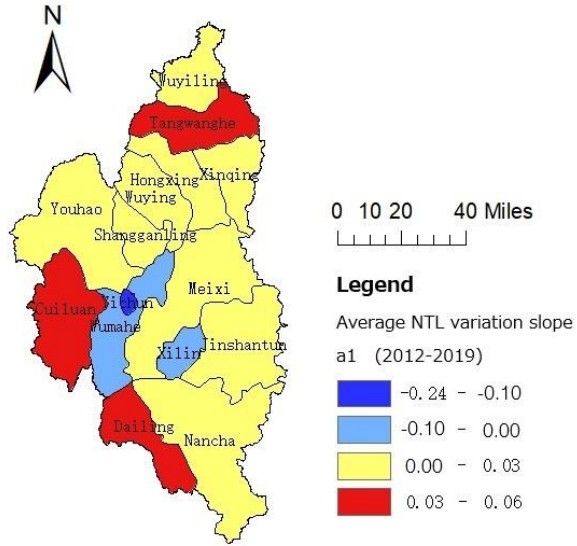

**Figure 4.** Slope distribution of average NTL changes in various districts of Yichun.

4.1.2. Partial Shrinkage

With scaling down, we quantified the mean value of the slope of the NTL radiance value to the sub-district level. It was found that the uneven development of Yichun City was not only reflected between the districts, but also in the interior of each district, with most of the districts showing partial shrinkage (Figure 5). From the perspective of the whole city, the partial shrinkage shows that the slope of the overall average NTL fluctuation in some districts is negative. From the inside of each district, the characteristic of partial shrinkage is that some towns/streets/forest farms in the districts are experiencing significant shrinkage while the district as a whole is in a state of stagnated development; these areas of shrinkage are mainly distributed in secondary blocks, old industrial areas, and scattered forest farms. According to the difference in the growth and shrinkage patterns of each district at the district and sub-district levels, the 15 districts of Yichun can be divided into the following five categories: growth district, stagnant and stable district, district where shrinkage and growth co-exist in a stable manner, partial light shrinkage district, and district of significant shrinkage. The classification results are shown in Table 1.

The classification results revealed that most of the streets in the district of significant shrinkage showed mild or significant shrinkage, especially in Yichun district, where most streets showed significant shrinkage. Hongxing District, Wuying District, and Youhao District were seen to be characterized by partial mild shrinkage, with the location of the shrinking parts of these three districts being secondary blocks and forest farms. Meixi District, Xinqing District, and Nancha District could generally be characterized by stagnant development, with the interior growth and shrinkage of their streets co-existing. Among them, both Xinqing District and Meixi District could be characterized by the growth of the main blocks and the shrinkage of the secondary blocks. The respective development of the north and south of Nancha District emerged as markedly different in that the shrinkage in the south was more significant, showing a concentrated shrinkage distribution, while the north mainly showed a growth trend.

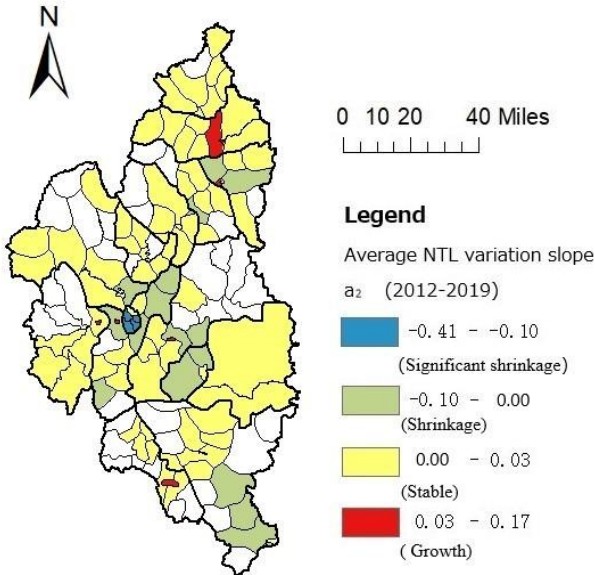

**Figure 5.** Average variation slope of NTLs on a sub-district scale in Yichun.

**Table 1.** Division of growth and shrinkage districts.

| Type | Criteria | | Feature Description |
| --- | --- | --- | --- |
| | District Level(a1) | Sub-District Level(a2) | |
| Growth district | a1 ≥ 0.03 | | Average slope of NTL radiance value in the district is increasing |
| Stagnant and Stable district | | 0 ≤ a2 < 0.03 | Average slope of NTL radiance value in the district and sub-district is stable |
| District where shrinkage and growth co-exist in a stable manner | 0 ≤ a1 < 0.03 | ∃ a2 ≥ 0.03 and a2 < 0 | Average slope of NTL radiance value in the district is stable, internal growth and shrinkage co-exist |
| Partial light shrinkage district | | ∃ a2 < 0 | Average slope of NTL radiance value in the district is stable, there exists partial shrinkage but no growth |
| District of significant shrinkage | a1 < 0 | | Average slope of NTL radiance value in the district is shrinking |

### 4.1.3. Changes in Regional Development Differences

From 2012 to 2019, the NTL radiance high value area in the urban area of Yichun City was mainly concentrated in Yichun District, with the NTL intensity of each district being generally low (Figure 6). The average NTL radiance value of most districts in 2018 was less than 3 nanoWatts/cm2/sr. The average NTL radiance value of Yichun District, the most economically prosperous and populous district, was only 10.97 nanoWatts/cm2/sr. However, at the same time, Nangang District, the district containing the highest NTL radiance value in Harbin, reached 41.06 nanoWatts/cm2/sr. Compared with the average NTL radiance values of various districts in Yichun in 2013, the NTL radiance value in Yichun District decreased significantly, while that in Tangwanghe District and Dailing District increased. The average NTL radiance value between the districts could thus be seen as small, with this gap having shrunk in recent years. The coefficient of variation of the average light intensity between the districts decreased from 1.51 in 2013 to 0.85 in 2018. This indicates a trend towards a low-level equilibrium between the various districts in Yichun City and a lack of core driving effects in the city' development.

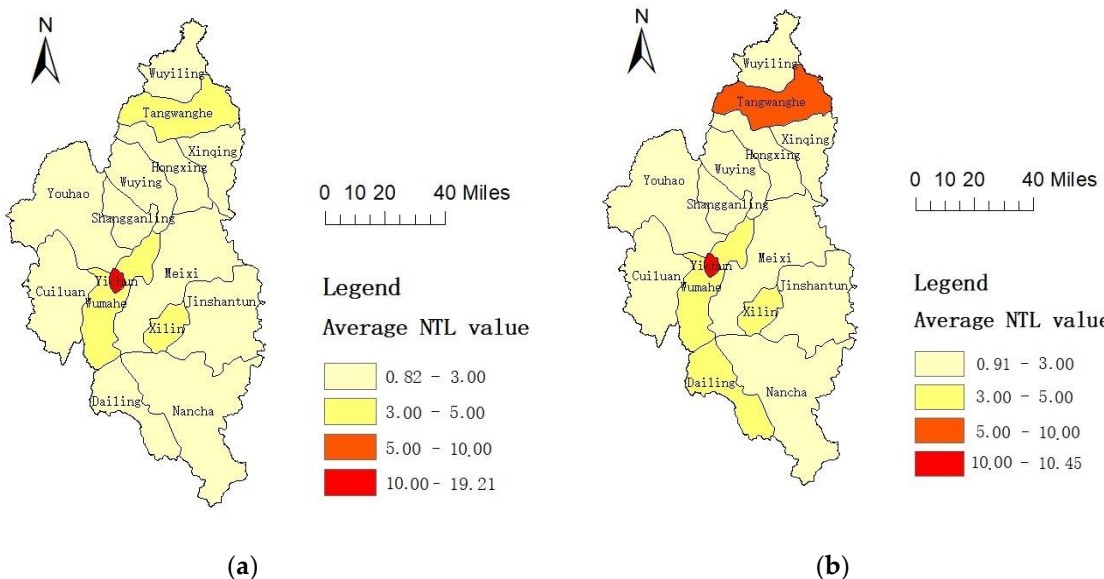

**Figure 6.** Average NTL radiance value in various areas of Yichun during (**a**) the year 2013; (**b**) the year 2018.

## 4.2. Refined Recognition

There are 19,252 grids of 100 m * 100 m in the built-up area of urban Yichun. Among these, there are 3913 grids with a shrinking monthly light change rate (20.32%), 9733 grids with a stable monthly light change (50.56%), and 5603 grids with growing monthly light change (28.52%). This result also certifies that the partial shrinkage in Yichun is significant. Here, we overlaid the NTL data and Google satellite imagery to more accurately identify the internal growth and shrinkage patterns of the urban built-up area in Yichun City. This overlay analysis proved more conducive to accurately understanding the spatial distribution of shrinkage and further exploring the cause of urban shrinkage.

Due to the scattered distribution of the built-up areas in forest farms, some of these areas are not shown in the overlay analysis demonstrated in Figure 6. The slope of the change in the NTL radiance value of forest farms mostly shows stagnant development, with some farms characterized by mild shrinkage. The reason for this is that the slope of these areas where the NTL value emerged as weak is also relatively small. However, according to the field investigation, the main functions of the existing forest farms are forest protection and multi-forest management. With the implementation of the shed renovation project, many forest farms have been withdrawn and merged, and the population of each forest farm has decreased sharply, making the shrinkage actually very significant. Therefore, we can conclude that the slope of the NTL radiance value is small, corresponding to the weak area of the NTL value, and that there is a phenomenon of the slope of the NTL change underestimating the shrinkage condition of the weak NTL value area.

### 4.2.1. Growth Area

The grid with a growing slope of the NTL radiance value was found to be distributed in most districts, with a very concentrated distribution in each district–mainly distributed in the core block. Among all districts, most of the built-up areas in the three growth districts showed a growing trend (Figure 7). Henan street in Tangwanghe District showed a significant growth trend, given that all of the grids in this street showed a growth trend and the grids of significant growth accounted for 36.21%. Regarding the other districts, while the distribution of the growth grids was also concentrated, the growth trend was not significant.

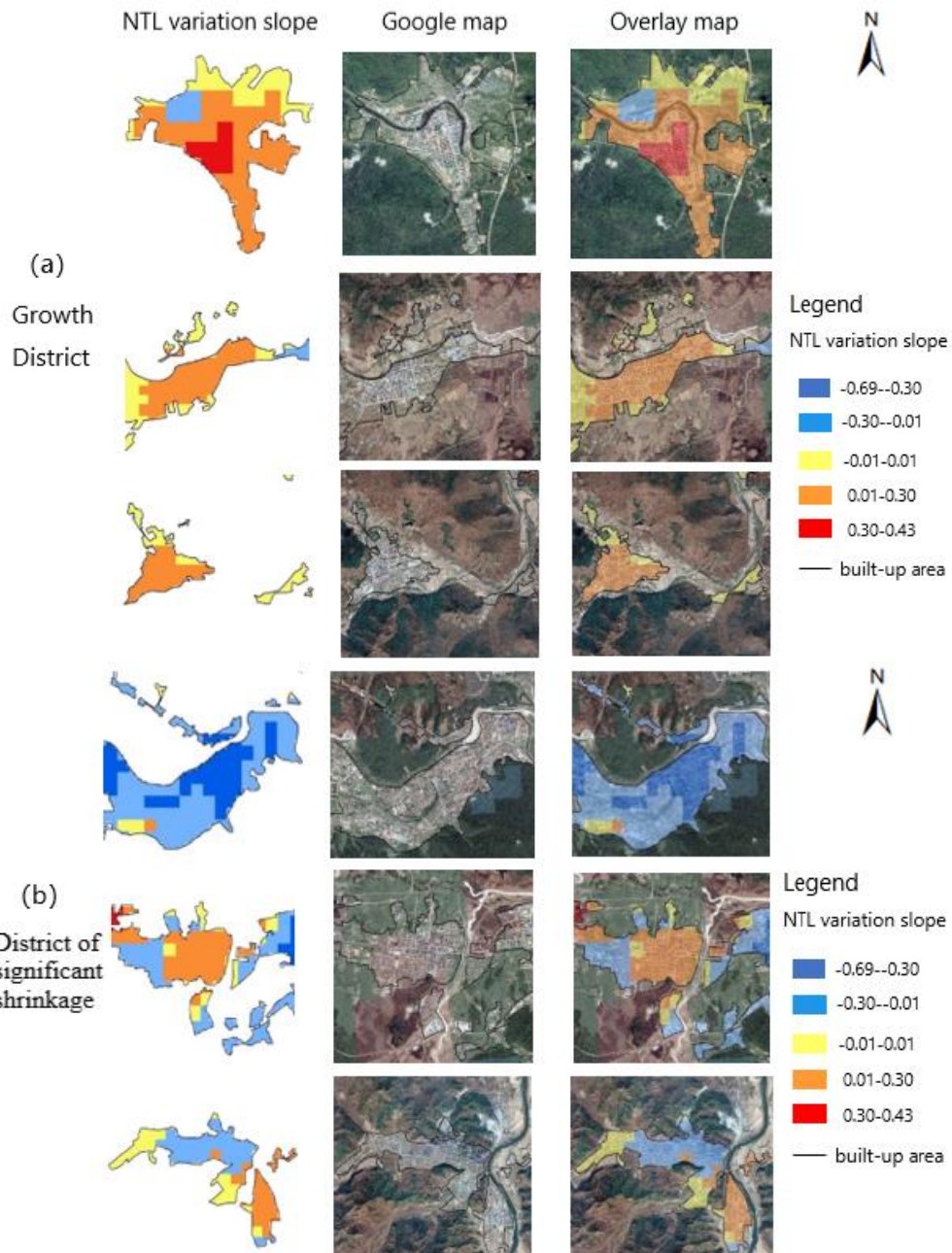

**Figure 7.** *Cont.*

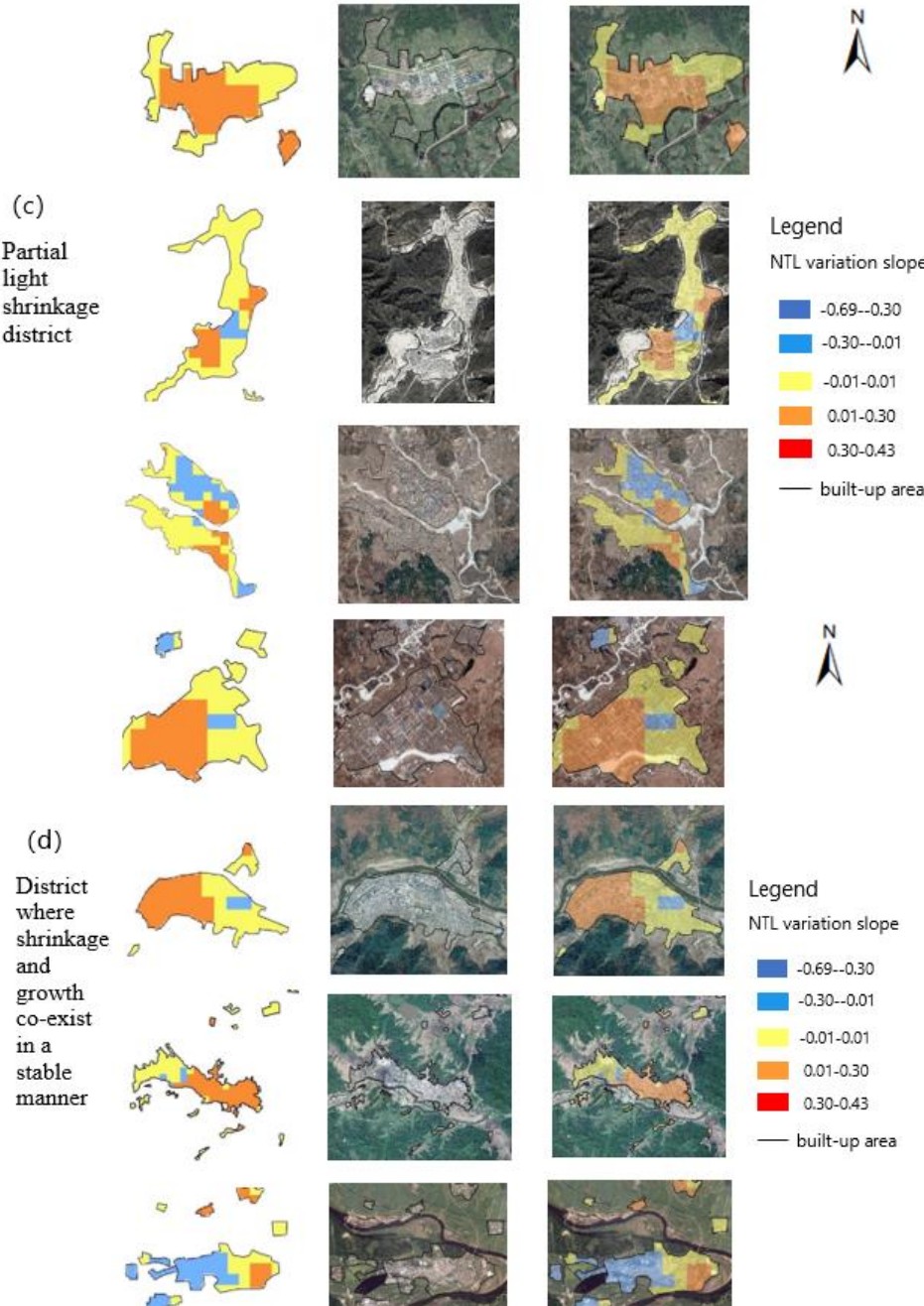

**Figure 7.** *Cont*.

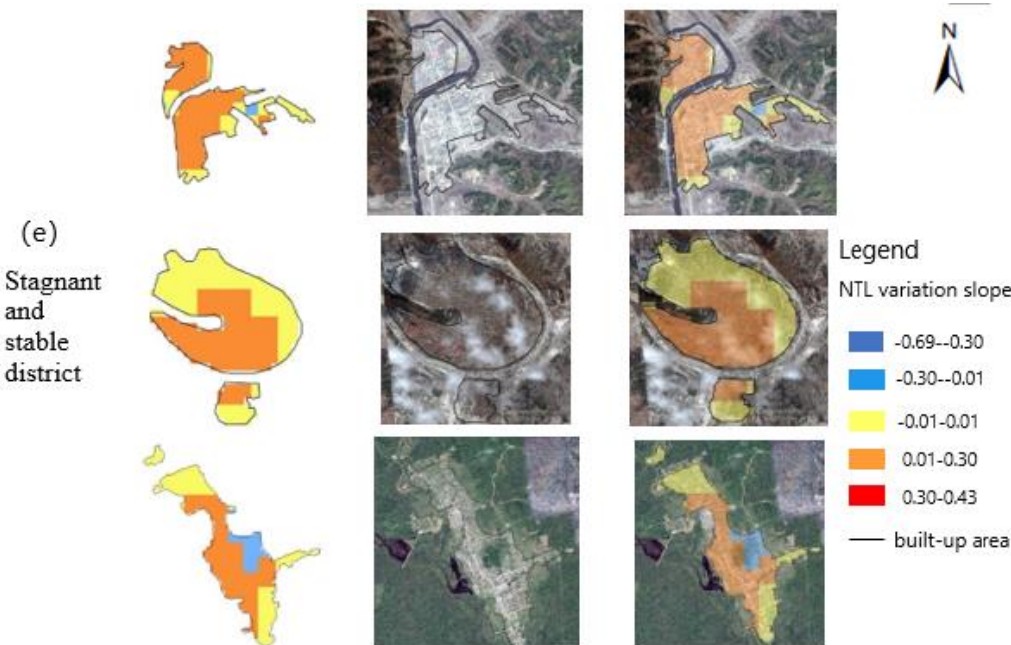

**Figure 7.** Shrinking and growing lights and land use of each district of Yichun.

Regional growth is not only due to pure market choices, but is also the result of active government interventions to optimize the spatial structure. By observing Google satellite imagery, the core block could be seen to be dominated by commercial and residential land. In recent years, the living environment of the core block has been improved and the supporting infrastructure updated. Part of the population of the forest farm and the secondary block has been concentrated in the core of each district. In addition, with the concern and support of the continuous and substitute industry, the Municipal Government in Yichun has been committed to the transformation of commercial blocks and the construction of economic zones. For example, the North Red Agate Jade Culture Industrial Base in Wuyiling District was built in 2016, which promoted the development of tourism in this block.

4.2.2. Shrinking Areas

Different degrees and scales of shrinkage were found in most districts of Yichun City, and the city was seen to have significant partial shrinkage. The distribution of the shrinkage grid inside the area mainly includes the built-up area of the core block, the forest farm, the old industrial area, and the townships far from the core built-up area. There are different reasons for the different shrinkage distributions (Table 2).

The shrinkage grids of Yichun District and Wuying District are mainly distributed in the core block. 1547 shrinkage grids were found in the 1665 100 m * 100 m grids in Yichun District, accounting for 92.9%, with 549 grids with significant shrinkage, accounting for 33%; the grids of significant shrinkage were mainly distributed throughout Hongsheng Street. According to the current research, a particular reason for the shrinkage of Yichun District is that the housing in Yichun District is in short supply, causing the predicament that the population in the region is being lost while the people of other districts cannot move in. The shrinkage grids of Xilin District emerged as concentrated in the northern part of Xinxing Street, with the industrial area mainly distributed in the north. With the implementation of the policy of eliminating backward production capacity, the Xilin Iron and Steel Industry reduced its production on a large scale, resulting in a significant shrinkage of the region. The shrinkage grids of Xinqing District, Meixi District, Youhao District, and Hongxing District were mainly found to be distributed in their forest farms owing to the policy ban on logging. The shrinking grids in Nancha District were mainly distributed in Haolianghe Town. Northern Cement and the Haolianghe Fertilizer Plant in this town, accounting for a high proportion of the industrial output value, both reduced

and stopped production in 2014, which caused a significant economic recession. The shrinkage grids in the Wumahe district were mainly distributed in the old industrial areas and the secondary school distribution areas of the secondary blocks. This could be seen as due to the gradual decline of industries in the old industrial areas, which caused the reduction of employment opportunities and the serious outflow of the young and middle-aged population, following which the school attendance population in secondary schools also gradually decreased.

**Table 2.** Shrinkage distribution and driving factors.

| Shrinkage Distribution | District | Driving Factors |
| --- | --- | --- |
| Core block | Yichun District, Wuying District | 1. Short housing supply<br>2. Low birth rate/migration<br>3. Service industry decline |
| Forest farm | Xinqing District, Meixi District, Youhao District, Hongxing District | 1. Policy ban on logging<br>2. Withdrawal and merging of forest farms and ecological migration |
| Old industrial area | Xilin District | 1. Diminished excess industrial capacities/reforms<br>2. Economic decline of sectors/jobs |
| Secondary block | Nancha District, Wumahe district | 1. Economic decline of sectors/jobs<br>2. Low birth rate/migration |

## 5. Discussion and Conclusions

### 5.1. Discussion

#### 5.1.1. The Characteristics of Shrinkage

The findings show a co-existence between the growth and shrinkage of Yichun, identifying three growth districts in Yichun, three stagnant and stable districts, three partial light shrinkage districts, three districts where shrinkage and growth co-existed in a stable manner, three partial light shrinkage districts, and three districts of significant shrinkage. At the district level, due to the comprehensive impact of resource endowments, industrial decline and policies, there exist unbalanced development trends among the various districts in Yichun, causing the differences between the regions to narrow. On the sub-district scale, most of the districts appeared to have different internal degrees and scales of shrinkage, with the areas of shrinkage concentrated in the secondary blocks and forest farms. On the grid scale, the grids of growth were mainly distributed in the core blocks of each district, while the grids of shrinkage were distributed in the built-up areas of the core blocks, forest farms, old industrial areas, and towns far from the core built-up areas.

#### 5.1.2. The Performance of NPP-VIIRS NTL Data

Due to the lack of data on the region's systematic population loss, the time series of the NPP-VIIRS NTL data were applied to identify the shrinkage pattern of the city given their advantage of timely capturing the dynamic changes of the ground source. In addition, given the high spatial resolution, the identification of urban shrinkage was possible, not only at the district and sub-district scales by way of the zonal statistic, but also at the grid scale of 100 m * 100 m. This attempt avoided the restrictions of administrative divisions and thus refined the identification of growth and shrinkage within the region.

This paper also combined high-resolution satellite imagery and socio-economic data in the process of using nighttime lighting data to identify urban shrinkage patterns. The severe population loss in the shrinking area shows that the NPP-VIIRS NTL data contain a certain objective reality in identifying urban shrinkage, which lays a foundation for further utilizing NTL data to identify urban shrinkage. In addition, given their greater degree of spatial detail, these data proved to be more applicable for accurately identifying the distribution of shrinkage, and exploring the shrinking reasons behind this, when overlaid with Google satellite imagery in the analysis.

Although the NPP-VIIRS NTL data here demonstrated their accuracy in the context of identifying urban shrinkage, the measure calculating the change slope of each grid's NTL radiance value still contained some uncertainties. The identification of the shrinkage pattern by using the slope change of the radiance value of the NTL underestimates the shrinkage of the area of weak NTL. The reason for this is that the slope of the calculated radiance change is also small for areas where the NTL radiance is weak, meaning that most of these areas are characterized as stagnant development and mild shrinkage areas, such as some forest farms and secondary blocks in Yichun. Through current field research, it was found that these areas with weak nightlights were facing serious population loss, and that the shrinkage was very significant.

## 5.2. Conclusions

Identification of city shrinkage is a precondition for the government and the policy makers to revitalize shrinking cities. To date, the methods of identifying urban shrinkage and growth have focused on traditional statistical methods, both in China and internationally. However, traditional population data has the defects of collection difficulties, inaccurate data statistics, statistical standard changes, long re-update cycles, and rough spatial expressions. What is more, the research scale has also been limited by administrative divisions, with few studies having refined the understanding of shrinkage and growth patterns within administrative boundaries. Thus, it is a major challenge to find objective time-series data to overcome these defects. Previous studies have proved that NTL radiance values have a significant statistical relationship with gross domestic product (GDP) and population. Thus NPP-VIIRS NTL data can be viewed as a combination of economic and demographic data that avoid the inaccuracy of traditional data statistics and the unilateral problem of using population data to identify urban shrinkage. In addition, the high spatial resolution of NPP-VIIRS NTL data makes it possible to refine the internal shrinkage of a city. Our study, therefore, fills a gap in this field to some degree.

In this paper, we made some meaningful attempts to refine the understanding of urban shrinkage. Firstly, we introduced Google satellite imagery and extracted the urban built-up area by visual interpretation, which made the study scope in line with international urban shrinkage research and solved the problem of blooming effect at the same time. Secondly, we used the latest NPP-VIIRS NTL data to identify the growth and shrinkage patterns through trending analysis, which provides a new case study for the extensive use of NTL data to explore urban shrinkage. In addition, combined with high-resolution Google satellite imagery and socio-economic data, we specified the shrinkage distribution and confirmed the objective reality of using NPP-VIIRS NTL data. Regarding urban managers, a key implication of this study is that it is only by accurately identifying the distribution of urban internal shrinkage and growth that regional development planning can be adequately targeted. Moreover, the case study in this paper demonstrated that urban shrinkage in China has the characteristic of local shrinkage, which is reflected at three spatial scales, namely, the district scale, the sub-district scale, and the grid scale, thus enriching the body of international research on the shrinking city.

For the further application of NTL data, the identification of shrinkage in weak areas needs further improvement for the trending analysis method underestimates the shrinkage of the area of weak NTL. At the same time, this result enlightens us to the necessity of undertaking fieldwork in order to understand the causes of urban light generation and change. In particular, regarding areas where NTL values are abnormal, the cause of the abnormality should be examined instead of directly smoothing or removing the outliers. Additionally, this paper takes a single city as an example. In future work, we will expand the scope of our research to the national scale in order to explore the law of distribution of shrinking cities.

**Author Contributions:** Conceptualization, C.L.; methodology, J.Z.; formal analysis, Y.Z.; investigation, Y.Z.; writing—original draft preparation, Y.Z.; writing—review and editing, Z.M.; visualization, S.H.; supervision, W.L.; funding acquisition, C.L.

**Funding:** This research was funded by National Natural Science Foundation (NSF) of China, grant number: 4187010342.

**Conflicts of Interest:** The authors declare no conflict of interest.

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
