# Peer review of "Identification of Shrinkage and Growth Patterns of a Shrinking City in China Based on Nighttime Light Data: A Case Study of Yichun"

_sustainability, doi:10.3390/su11246906_

Round 1

Reviewer 1 Report

This paper uses a trending method and does some interesting works. Overall it is well-structured. I find the major weakness is that it does not make a very good theoretical review and discussions, so the paper looks more like case study rather than an original contribution. Also, some result presentations are not satisfactory and need improvement. Please see detailed comments below:

Line 12-13: This is not accurate, as you can find many big data and shrinking cities studies now.

Line 51: “Urban space science” is not a well-used terminology. It might be urban geography or geospatial urban studies.

Introduction: The introduction focuses too much on VIIRS technical discussions, which is not a good focal point because the paper does not actually contribute too much to the method or the data itself. An application of VIIRS is fine, but the authors need to invest more efforts into explaining why their works have a good practical contribution with the dataset and review previous contributions, such as:

Does this study lead to planning support tools or methods?

https://link.springer.com/article/10.1007/s12061-019-09296-5

https://journals.sagepub.com/doi/full/10.1177/2399808317743971

Does it contribute to big data/shrinking city studies?

https://www.tandfonline.com/doi/abs/10.1080/19479832.2016.1215355

Line 137: I think this paper devotes too many discussions to a local site rather than the big picture of how this case contributes to the larger academia of shrinking cities.

Line 161: In the data pre-processing part, the authors should first list all the data challenges (such as blooming, inter-period inconsistencies) and what kind of time-series your study needs, then dive into each pieces of pre-processing. Now it is hard to grasp why all these parts contribute to a coherent dataset.

Figure 4: In general, the figures in this study are too small and the legends do not have consistent decimals (I think the authors should use 2 consistently). The characters are very hard to see in the figures. Most international readers will not be interested in what these districts are called. I would suggest other forms of notifications with more insightful information.

Table 1: Again, I really do not think international readers will be interested in district names. Some spatial patterns will be more useful.

Figure 6: This figure looks good and interesting. However, the authors should not use the district names, but rather group these figures into typologies and patterns.

Discussions: This paper needs to add a separate discussion to show the contribution of this study, especially to the larger academia, not just this case. For example, how this study compares to the urban growth/shrinkage study with land use data or socioeconomic data?

https://www.sciencedirect.com/science/article/pii/S0959652619318104

Author Response

Revision details to the Manuscript ID: sustainability-643189

Title: Identification of Shrinkage and Growth Patterns of a Shrinking City in China Based on NPP-VIIRS NTL Data: A Case Study of Yichun

Dear reviewer,

Thank you for your constructive comments and suggestions for this article, which have greatly helped us to further improve the article. We have considered the comments very seriously and revised the manuscript accordingly.

The replies are as follows:

Response to Reviewer 1 Comments

Point 1: Line 12-13: This is not accurate, as you can find many big data and shrinking cities studies now. 

Response 1: Although there are some existing studies that are based on big data, they are lack of time series data, the identification of shrinking cities mostly relied on traditional socio-economic data. So, to make it more accurate,  we replaced the sentence “ However, the methods of identifying urban shrinkage and growth have been limited to traditional statistical methods”to “However, the methods of identifying urban shrinkage and growth have mostly focused on traditional statistical methods and studies based on NTL data are rare .

Point 2: Line 51: “Urban space science” is not a well-used terminology. It might be urban geography or geospatial urban studies.

Response 2: We replaced the terminology “Urban space science” with “Urban geography”

Point 3: Introduction: The introduction focuses too much on VIIRS technical discussions, which is not a good focal point because the paper does not actually contribute too much to the method or the data itself. An application of VIIRS is fine, but the authors need to invest more efforts into explaining why their works have a good practical contribution with the dataset and review previous contributions, such as:

Does this study lead to planning support tools or methods?

https://link.springer.com/article/10.1007/s12061-019-09296-5

https://journals.sagepub.com/doi/full/10.1177/2399808317743971

Does it contribute to big data/shrinking city studies?

https://www.tandfonline.com/doi/abs/10.1080/19479832.2016.1215355

Response 3: With reference to your suggestions and the references you provided, we have added the application of NPP-VIIRS NTL data in the introduction and the implications for this article.

To date, urban studies, using NPP-VIIRS NTL data, have focused on the following aspects: (1) the relationships between the NPP-VIIRS NTL data and socioeconomic indicators, such as population, GDP, and house vacancy rate; (Li et al. 2013; Shi et al. 2014; Ma et al. 2014; Chen et al. 2015); (2) extracting built-up areas and dynamically monitoring urban expansion by the NPP-VIIRS NTL data (Shi et al. 2014; Miles et al. 2014; Yu et al. 2014); (3) exploring the spatial distribution of the population using NTL data (Xie et al. 2014; Hu et al. 2014;). These studies have provided insights and foundations for using NPP-VIIRS NTL data to identify urban shrinkage and growth patterns. Firstly, many studies have shown that NTL radiance values have a significant statistical relationship with gross domestic product (GDP) and population, which illustrates the comprehensiveness of NTL data and its strong relevance to human activities. In addition, the capacity of NTL time-series data to dynamically detect changes in urban landscapes demonstrates the objectivity and timeliness of these data. Finally, using NPP-VIIRS NTL data to spatialize socio-economic indicators can be seen as more reliable than using traditional data. In order to test the superiority of NPP-VIIRS NTL data,some scholars have made meaningful attempts to contribute to research on city shrinkage. At present, there are two methods of processing the NTL data for urban shrinkage research. Firstly, Du et al. (2017) and Liu et al. (2018) employed a calculation of the difference of different years’ NTL radiance value in every grid to identify urban shrinkage and growth; however, this difference method cannot determine the continuity and trend of urban shrinkage and growth. Secondly, Li et al. (2019) used the NPP/VIIRS NTL data in the calculation of the change slope of each grid’s NTL radiance value to identify the urban shrinkage and growth pattern; however, the data here were not carefully processed through the implementation of noise removal. Moreover, the scope of the research focused on the whole prefecture-level city rather than the urban area, which differs from the western cities where there are urbanized areas within administrative boundaries.

Point 4: Line 137: I think this paper devotes too many discussions to a local site rather than the big picture of how this case contributes to the larger academia of shrinking cities.

Response 4: This paper identifies the shrinkage patterns within the city from three scales in order to provide an analytical framework to other scholars who want to use nighttime light data to identify urban shrinkage. In response to your question, we highlighted our contributions by adding an independent discussion section and explaining the significance of our work.

Point 5: Line 161: In the data pre-processing part, the authors should first list all the data challenges (such as blooming, inter-period inconsistencies) and what kind of time-series your study needs, then dive into each pieces of pre-processing. Now it is hard to grasp why all these parts contribute to a coherent dataset.

Response 5: With on-board correction and satellite correction, the inter-annual systematic geolocation shift of NPP-VIIRS NTL data is small, so we assume that the monthly data for 2012-2019 is a coherent dataset. And the choose of data, we have showed in the part of data source. To your suggestion, we have listed the challenges and study needs before pre-processing.

Given the characteristics of NPP-VIIRS time series images and the needs of specific research problems, the images need to be corrected prior to undertaking the analysis. With on-board correction and satellite correction, the inter-annual systematic geolocation shift of NPP-VIIRS NTL data is small; for this reason, so we assumed that the monthly data for 2012-2019 comprised a coherent dataset. We then removed extreme values from the image by determining the optimal threshold. In addition, due to the characteristics of the artificial nighttime radiation source and the sensor itself, the existence of the temporary light source and the blooming effect are two important phenomena to consider when using NPP-VIIRS data. To rectify these defects, we attempted to perform specific processing by using high-resolution Google satellite imagery.

Point 6: Figure 4: In general, the figures in this study are too small and the legends do not have consistent decimals (I think the authors should use 2 consistently). The characters are very hard to see in the figures. Most international readers will not be interested in what these districts are called. I would suggest other forms of notifications with more insightful information.

Response 6: We calculated the monthly variation slope of the nighttime light value of each grid, and then averaged them to each administrative unit, so the value looked small.

Figure 4. Average variation slope of NTLs on sub-district scale in Yichun

Point 7: Table 1: Again, I really do not think international readers will be interested in district names. Some spatial patterns will be more useful.

Response 7:

Table 1. Division of growth and shrinkage districts

Type

Criteria

District level a1

Sub-district level

a2

Feature description

Growth district

a1≥0.03

Average slope of NTL radiance value in district is increasing

Stagnant and Stable district

0≤a1<0.03

0≤a2<0.03

Average slope of NTL radiance value in district and sub-district is stable

District where shrinkage and growth co-exist in a stable manner

ョa2≥0.03 & a2<0

Average slope of NTL radiance value in district is stable, internal growth and shrinkage co-exist

Partial light shrinkage district

ョa2<0

Average slope of NTL radiance value in district is stable, there exists partial shrinkage but no growth

District of significant shrinkage

a1<0

Average slope of NTL radiance value in district is shrinking

Point 8: Figure 6: This figure looks good and interesting. However, the authors should not use the district names, but rather group these figures into typologies and patterns.

Response 8:

Figure 6. Shrinking and growing lights and land use of each district of Yichun

Point 9: Discussions: This paper needs to add a separate discussion to show the contribution of this study, especially to the larger academia, not just this case. For example, how this study compares to the urban growth/shrinkage study with land use data or socioeconomic data?

Response 9:

5 Discussion and Conclusions

5.1 Discussion

5.1.1. The characteristics of shrinkage

The findings show a co-existence between the growth and shrinkage of Yichun, identifying three growth districts in Yichun, three stagnant and stable districts, three partial light shrinkage districts, three districts where shrinkage and growth co-existed in a stable manner, three partial light shrinkage districts, and three districts of significant shrinkage. At the district level, due to the comprehensive impact of resource endowments, industrial decline and policies, there exist unbalanced development trends among the various districts in Yichun, causing the differences between the regions to narrow. On the sub-district scale, most of the districts appeared to have different internal degrees and scales of shrinkage, with the areas of shrinkage concentrated in the secondary blocks and forest farms. On the grid scale, the grids of growth were mainly distributed in the core blocks of each district, while the grids of shrinkage were distributed in the built-up areas of the core blocks, forest farms, old industrial areas and towns far from the core built-up areas.5.1.2. The performance of NPP-VIIRS NTL data

5.1.2. The performance of NPP-VIIRS NTL data

Due to the lack of data on the region’s systematic population loss, the time series of the NPP-VIIRS NTL data were applied to identify the shrinkage pattern of the city given their advantage of timely capturing the dynamic changes of the ground source. In addition, given the high spatial resolution, the identification of urban shrinkage was possible, not only at the district and sub-district scales by way of the zonal statistic, but also at the grid scale of 100m*100m. This attempt avoided the restrictions of administrative divisions and thus refined the identification of growth and shrinkage within the region.

This paper also combined high-resolution satellite imagery and socio-economic data in the process of using nighttime lighting data to identify urban shrinkage patterns. The severe population loss in the shrinking area shows that the NPP-VIIRS NTL data contain a certain objective reality in identifying urban shrinkage, which lays a foundation for further utilizing NTL data to identify urban shrinkage. In addition, given their greater degree of spatial detail, these data proved to be more applicable for accurately identifying the distribution of shrinkage, and exploring the shrinking reasons behind this, when overlaid with Google satellite imagery in the analysis.

Although the NPP-VIIRS NTL data here demonstrated their accuracy in the context of identifying urban shrinkage, the measure calculating the change slope of each grid’s NTL radiance value still contained some uncertainties. The identification of the shrinkage pattern by using the slope change of the radiance value of the NTL underestimates the shrinkage of the area of weak NTL. The reason for this is that the slope of the calculated radiance change is also small for areas where the NTL radiance is weak, meaning that most of these areas are characterized as stagnant development and mild shrinkage, such as some forest farms and secondary blocks in Yichun. Through current field research, it was found that these areas with weak nightlights were facing serious population loss, and that the shrinkage was very significant.

5.2 Conclusions

5.2.1. Contribution to scholarship

To date, the methods of identifying urban shrinkage and growth have focused on traditional statistical methods, both in China and internationally. The scale of this research scale has also been limited by administrative divisions, with few studies having refined the understanding of shrinkage and growth patterns within administrative boundaries. This paper uses the latest NPP-VIIRS NTL data to identify the growth and shrinkage patterns at the 100*100 grid scale in Yichun, combined with high-resolution Google satellite imagery and socio-economic data, thus making a meaningful attempt to refine the aforementioned understanding. NTL data can be viewed as a combination of economic and demographic data that avoid the inaccuracy of traditional data statistics and the unilateral problem of using urban population data to identify urban shrinkage. Our study, therefore, fills a gap in this area.

5.2.2. Contribution to practice

This paper provides a way to refine the identification of urban shrinkage by using NPP-VIIRS NTL data combined with urban remote sensing imagery and socio-economic data. This provides a new perspective for the extensive use of NTL data in urban shrinkage case studies. In addition, the case study in this paper demonstrated that urban shrinkage in China has the characteristic of local shrinkage, which characteristic is reflected at three spatial scales, namely, the district scale, the sub-district scale, and the grid scale, thus enriching the body of international research on the shrinking city. Moreover, regarding urban managers, a key implication of this study is that it is only by accurately identifying the distribution of urban internal shrinkage and growth that regional development planning can be adequately targeted.

5.2.3. Future work

For the further application of NTL data, the identification of shrinkage in weak areas needs further improvement. At the same time, this result enlightens us to the necessity of undertaking fieldwork in order to understand the causes of urban light generation and change. In particular, regarding areas where NTL values are abnormal, the cause of the abnormality should be examined instead of directly smoothing or removing the outliers. Additionally, this paper takes a single city as an example, which is of small research scope. In future work, we will expand the scope of our research to the national scale in order to explore the law of distribution of shrinking cities.

Reviewer 2 Report

The paper titled “Identification of Shrinkage and Growth Patterns of a Shrinking City in China Based on NPP-VIIRS NTL Data: A Case Study of Yichun” presents an interesting and original point of view on the methodology to evaluate the quality and quantity of shrinkage in different cities.

The paper is coherently organized, the title is self-explanatory also if it seems too long, the abstract specify directly the aim of the paper, and the tables and figures are satisfactory from the point of view of the data collection and analysis. In the opinion of the writer, the results can be also erased from the abstract.

Overall, the entire paper is an interesting, original and coherent product of research, with minor issues related to minor wording phrases and a secondary lack in the bibliography, especially in paragraphs 4 and 5.

Can be useful to introduce the study area (Yichun) in a broader Asian or Chinese context, underlying differences and similarities between western and eastern cases that, currently, are generically described between Rows 92 and 98.

The choice of the site, especially related to the phenomenon of logging reduction, is coherent with the paper, and also the contraction in the number of inhabitants make the city an interesting case study. Probably can be useful a paragraph to answer to the following question: can the results of the research be interesting and useful for a general readership interested in urban changes?

Conclusion section should provide general conclusions and the presented arguments should be supported by active usage of references. What are the consequences of your research for management of regions and cities impacted by urban shrinking?

In general, the theme and methodology are of sure interest to both a wide audience and a specialist, especially for the use of an innovative approach. Overall, the entire paper is an interesting, original and coherent.

Conclusions and Discussion are appropriated and balanced, also if a better use of references can be useful to put the paper in a broader scientific context. Bibliography is appropriate. Overall, the paper is clear and easy to understand. The results are interesting with good potential for future researches.

Hope my comments will be useful to the authors.

In general, the theme and methodology are of sure interest to both a wide audience and a specialist, especially for the use of an innovative approach. Overall, the entire paper is an interesting, original and coherent.

Conclusions and Discussion are appropriated and balanced, also if a better use of references can be useful to put the paper in a broader scientific context. Bibliography is appropriate. Overall, the paper is clear and easy to understand. The results are interesting with good potential for future researches.

Hope my comments will be useful to the authors.

Author Response

Revision details to the Manuscript ID: sustainability-643189

Title: Identification of Shrinkage and Growth Patterns of a Shrinking City in China Based on NPP-VIIRS NTL Data: A Case Study of Yichun

Dear reviewer,

Thank you for your constructive comments and suggestions for this article, which have greatly helped us to further improve the article. We have considered the comments very seriously and revised the manuscript accordingly.

The replies are as follows:

Response to Reviewer 2 Comments

Point 1: The paper is coherently organized, the title is self-explanatory also if it seems too long, the abstract specify directly the aim of the paper, and the tables and figures are satisfactory from the point of view of the data collection and analysis. In the opinion of the writer, the results can be also erased from the abstract.

Response 1: According to your suggestion, we erased the results and highlighted the academic contributions of this paper in the abstract.

Abstract: Urban shrinkage has become a topic of great concern to scholars of geography and urban science. However, the methods of identifying urban shrinkage and growth have mostly focused on traditional statistical methods, and studies based on nighttime light (NTL) data are rare. Here, we use the NPP-VIIRS NTL data of 56 months from 2012 to 2019 to identify the shrinkage and growth patterns of Yichun in China, by calculating the slope of the NTL radiance value after denoising. At the same time, by combining high-resolution Google satellite images and traditional demographic data, we analyzed the shrinkage characteristics of Yichun. The results of the study confirmed the characteristics of partial shrinkage in China's shrinking cities. In addition, the use of NPP-VIIRS NTL data was able more accurately to identify the urban shrinkage and growth patterns, and may also be seen to present a more objective reality, thus providing a new perspective for studies of urban shrinkage.

Point 2: Overall, the entire paper is an interesting, original and coherent product of research, with minor issues related to minor wording phrases and a secondary lack in the bibliography, especially in paragraphs 4 and 5.

Response 2: In order to enrich the literature review, we added more reference in these sections.

Point 3: Can be useful to introduce the study area (Yichun) in a broader Asian or Chinese context, underlying differences and similarities between western and eastern cases that, currently, are generically described between Rows 92 and 98.

Response 3: According to your suggestion, we add the Chinese context to introduce the study area (Yichun).

In recent years, an increasing number of Chinese scholars has paid attention to shrinking cities, with a large number of studies having identified the situation and distribution of urban shrinkage in China [43-45]. These studies all indicate that urban shrinkage is common in China. For example, Long et al. (2015) used the 2000 and 2010 census data to define 180 shrinking cities in China; Wu et al. (2015) used the urban population data in 2007 and 2016 to identify 80 shrinking cities in China; Zhang et al. (2016) and Wu et al. (2018) found over a third of cities in China to be experiencing different degrees of population shrinkage at the county scale. Moreover, these results all showed that many cities in northeast China are experiencing the most serious shrinkage, and most of them are resource-based cities.

Point 4: The choice of the site, especially related to the phenomenon of logging reduction, is coherent with the paper, and also the contraction in the number of inhabitants make the city an interesting case study. Probably can be useful a paragraph to answer to the following question: can the results of the research be interesting and useful for a general readership interested in urban changes?

Response 4: This case study confirms the characteristics of partial shrinkage rather than overall recession in China's shrinking cities, which is different from the western cities. We displayed the academic significances of this paper in the discussion section.

5 Discussion and Conclusions

5.1 Discussion

5.1.1. The characteristics of shrinkage

The findings show a co-existence between the growth and shrinkage of Yichun, identifying three growth districts in Yichun, three stagnant and stable districts, three partial light shrinkage districts, three districts where shrinkage and growth co-existed in a stable manner, three partial light shrinkage districts, and three districts of significant shrinkage. At the district level, due to the comprehensive impact of resource endowments, industrial decline and policies, there exist unbalanced development trends among the various districts in Yichun, causing the differences between the regions to narrow. On the sub-district scale, most of the districts appeared to have different internal degrees and scales of shrinkage, with the areas of shrinkage concentrated in the secondary blocks and forest farms. On the grid scale, the grids of growth were mainly distributed in the core blocks of each district, while the grids of shrinkage were distributed in the built-up areas of the core blocks, forest farms, old industrial areas and towns far from the core built-up areas.5.1.2. The performance of NPP-VIIRS NTL data

5.1.2. The performance of NPP-VIIRS NTL data

Due to the lack of data on the region’s systematic population loss, the time series of the NPP-VIIRS NTL data were applied to identify the shrinkage pattern of the city given their advantage of timely capturing the dynamic changes of the ground source. In addition, given the high spatial resolution, the identification of urban shrinkage was possible, not only at the district and sub-district scales by way of the zonal statistic, but also at the grid scale of 100m*100m. This attempt avoided the restrictions of administrative divisions and thus refined the identification of growth and shrinkage within the region.

This paper also combined high-resolution satellite imagery and socio-economic data in the process of using nighttime lighting data to identify urban shrinkage patterns. The severe population loss in the shrinking area shows that the NPP-VIIRS NTL data contain a certain objective reality in identifying urban shrinkage, which lays a foundation for further utilizing NTL data to identify urban shrinkage. In addition, given their greater degree of spatial detail, these data proved to be more applicable for accurately identifying the distribution of shrinkage, and exploring the shrinking reasons behind this, when overlaid with Google satellite imagery in the analysis.

Although the NPP-VIIRS NTL data here demonstrated their accuracy in the context of identifying urban shrinkage, the measure calculating the change slope of each grid’s NTL radiance value still contained some uncertainties. The identification of the shrinkage pattern by using the slope change of the radiance value of the NTL underestimates the shrinkage of the area of weak NTL. The reason for this is that the slope of the calculated radiance change is also small for areas where the NTL radiance is weak, meaning that most of these areas are characterized as stagnant development and mild shrinkage, such as some forest farms and secondary blocks in Yichun. Through current field research, it was found that these areas with weak nightlights were facing serious population loss, and that the shrinkage was very significant.

Point 5: Conclusion section should provide general conclusions and the presented arguments should be supported by active usage of references. What are the consequences of your research for management of regions and cities impacted by urban shrinking?

Response 5:

5.2 Conclusions

5.2.1. Contribution to scholarship

To date, the methods of identifying urban shrinkage and growth have focused on traditional statistical methods, both in China and internationally. The scale of this research scale has also been limited by administrative divisions, with few studies having refined the understanding of shrinkage and growth patterns within administrative boundaries. This paper uses the latest NPP-VIIRS NTL data to identify the growth and shrinkage patterns at the 100*100 grid scale in Yichun, combined with high-resolution Google satellite imagery and socio-economic data, thus making a meaningful attempt to refine the aforementioned understanding. NTL data can be viewed as a combination of economic and demographic data that avoid the inaccuracy of traditional data statistics and the unilateral problem of using urban population data to identify urban shrinkage. Our study, therefore, fills a gap in this area.

5.2.2. Contribution to practice

This paper provides a way to refine the identification of urban shrinkage by using NPP-VIIRS NTL data combined with urban remote sensing imagery and socio-economic data. This provides a new perspective for the extensive use of NTL data in urban shrinkage case studies. In addition, the case study in this paper demonstrated that urban shrinkage in China has the characteristic of local shrinkage, which characteristic is reflected at three spatial scales, namely, the district scale, the sub-district scale, and the grid scale, thus enriching the body of international research on the shrinking city. Moreover, regarding urban managers, a key implication of this study is that it is only by accurately identifying the distribution of urban internal shrinkage and growth that regional development planning can be adequately targeted.

5.2.3. Future work

For the further application of NTL data, the identification of shrinkage in weak areas needs further improvement. At the same time, this result enlightens us to the necessity of undertaking fieldwork in order to understand the causes of urban light generation and change. In particular, regarding areas where NTL values are abnormal, the cause of the abnormality should be examined instead of directly smoothing or removing the outliers. Additionally, this paper takes a single city as an example, which is of small research scope. In future work, we will expand the scope of our research to the national scale in order to explore the law of distribution of shrinking cities.

Round 2

Reviewer 1 Report

In general the revision address the reviewer comments, but after another reading, I still find the methodology and result sections not clear and informative so additions are still necessary:

There is not a methodological flow so it is hard to read the flow of the method; Figure 3,4,6 still have inconsistent decimal numbers in the legend Original night-light imagery is preferred to show the original data of the paper; Section 3.2.1 does not jus address extreme values but other inconsistencies of the night-time imagery. It is good to use subsction for the list of changes to data inconsistencies and provide equations for the specific work you have done. 

Author Response

Revision details to the Manuscript ID: sustainability-643189

Title: Identification of Shrinkage and Growth Patterns of a Shrinking City in China Based on NPP-VIIRS NTL Data: A Case Study of Yichun

Dear reviewer,

Thank you very much for your constructive comments and suggestions for this article, which have greatly helped us to further improve the article. We have considered the comments very seriously and revised the manuscript accordingly.

The replies are as follows:

Response to Reviewer 1 Comments

Point 1: There is not a methodological flow so it is hard to read the flow of the method

Response 1:  With reference to your suggestion, we added the flowchart in section 3.2.

Figure 3. Flowchart of NPP-VIIRS NTL data processing

Point 2: Figure 3,4,6 still have inconsistent decimal numbers in the legend Original night-light imagery is preferred to show the original data of the paper

Response 2: We kept two decimal numbers in Fig4,5,6,7,  as follows:

Figure 4. Slope distribution of average NTL changes in various districts of Yichun

Figure 5. Average variation slope of NTLs on sub-district scale in Yichun

(b)

Figure 6. Average NTL radiance value in various areas of Yichun (a)year 2013; (b) year 2018.

Figure 7. Shrinking and growing lights and land use of each district of Yichun

Point 3: Section 3.2.1 does not jus address extreme values but other inconsistencies of the night-time imagery. It is good to use subsction for the list of changes to data inconsistencies and provide equations for the specific work you have done.

(1)

Response 3: According to your suggestions, we provided equation (1) in section 3.2.1 to make our method clearer.
